# Two applications of Min-Max-Jump distance

## Abstract

We explore two applications of Min-Max-Jump distance (MMJ distance): MMJ-based K-means and MMJ-based internal clustering evaluation index. K-means and its variants are possibly the most popular clustering approach. A key drawback of K-means is that it cannot deal with data sets that are not the union of well-separated, spherical clusters. MMJ-based K-means proposed in this paper overcomes this demerit of K-means, so that it can handle irregularly shaped clusters. Evaluation (or "validation") of clustering results is fundamental to clustering and thus to machine learning. Popular internal clustering evaluation indices like Silhouette coefficient, Davies–Bouldin index, and Calinski-Harabasz index performs poorly in evaluating irregularly shaped clusters. MMJ-based internal clustering evaluation index uses MMJ distance and Semantic Center of Mass (SCOM) to revise the indices, so that it can evaluate irregularly shaped data. An experiment shows introducing MMJ distance to internal clustering evaluation index, can systematically improve the performance. We also devise two algorithms for calculating MMJ distance.

## 1 Introduction

Distance is a numerical measurement of how far apart objects or points are. It is usually formalized in mathematics using the notion of a metric space. A metric space is a set together with a notion of distance between its elements, usually called points. The distance is measured by a function called a metric or distance function. Metric spaces are the most general setting for studying many of the concepts of mathematical analysis and geometry.

In this paper, we introduce two algorithms for calculating Min-Max-Jump distance (MMJ distance) and explore two applications of it. Including MMJ-based K-means (MMJ-K-means) and MMJ-based internal clustering evaluation index.

MMJ-K-means improves K-means, so that it can handle irregularly shaped clusters. We claim MMJ-CH is the SOTA (state-of-the-art) internal clustering evaluation index, which achieves an accuracy of $90/145$. MMJ-CH is one of the MMJ-based internal clustering evaluation indices.

## 2 RELATED WORK

### 2.1 Different distance metrics

Many distance measures have been proposed in literature, such as Euclidean distance or cosine similarity. These distance measures often be found in algorithms like k-NN, UMAP, HDBSCAN, etc. The most common metric is Euclidean distance. Cosine similarity is often used as a way to counteract Euclidean distance's problem in high dimensionality. The cosine similarity is the cosine of the angle between two vectors.

Submitted to 38th Conference on Neural Information Processing Systems (NeurIPS 2024). Do not distribute.

Hamming distance is the number of values that are different between two vectors. It is typically used to compare two binary strings of equal length (1).

Manhattan distance is a geometry whose usual distance function or metric of Euclidean geometry is replaced by a new metric in which the distance between two points is the sum of the absolute differences of their Cartesian coordinates (2).

Chebyshev distance is defined as the greatest of difference between two vectors along any coordinate dimension (3).

Minkowski distance or Minkowski metric is a metric in a normed vector space which can be considered as a generalization of both the Euclidean distance and the Manhattan distance (4).

Jaccard index, also known as the Jaccard similarity coefficient, is a statistic used for gauging the similarity and diversity of sample sets (5).

Haversine distance is the distance between two points on a sphere given their longitudes and latitudes. It is similar to Euclidean distance in that it calculates the shortest path between two points. The main difference is that there is no straight line, since the assumption is that the two points are on a sphere (6).

## 2.2 K-means

K-means (7) and its variants (8; 9; 10) are possibly the most well-liked clustering approach. K-means divides the data into K groups, where K is a hyper-parameter to be optimized. It aims to reduce the within-cluster dissimilarity. While popular, K-means and its variants perform poorly for data sets that are not the union of well-separated, spherical clusters. MMJ-based K-means (MMJ-K-means) proposed in this paper overcomes this demerit of K-means, so that it can handle irregularly shaped clusters.

## 2.3 Internal clustering evaluation index

Evaluation (or "validation") of clustering results is as difficult as the clustering itself (11). Popular approaches involve "internal" evaluation and "external" evaluation. In internal evaluation, a clustering result is evaluated based on the data that was clustered itself. Popular internal evaluation indices are Davies-Bouldin index (12), Silhouette coefficient (13), Dunn index (14), and Calinski-Harabasz index (15) etc. In external evaluation, the clustering result is compared to an existing "ground truth" classification, such as the Rand index (16). However, knowledge of the ground truth classes is almost never available in practice.

In Section 5.2, an experiment shows introducing Min-Max-Jump (MMJ) distance to internal clustering evaluation index, can systematically improve the performance.

## 2.4 Path-based distances

Euclidean distances are frequently used in machine learning and clustering methods to compare points. However, the distance is data-independent, and not tailored to the geometry of the data. Many metrics that are data-dependent have been devised, such as diffusion distances (17) and path-based distances (18; 19). MMJ distance is a path-based distance.

# 3 Definition of Min-Max-Jump

**Definition 1.** *Min-Max-Jump distance (MMJ distance)*

*$\Omega$ is a set of points (at least one). For any pair of points $p, q \in \Omega$, the distance between $p$ and $q$ is defined by a distance function d(p,q) (such as Euclidean distance). $i, j \in \Omega$, $\Psi_{(i,j,n,\Omega)}$ is a path from point i to point j, which has length of n points (see Table 1). $\Theta_{(i,j,\Omega)}$ is the set of all paths from point i to point j. Therefore, $\Psi_{(i,j,n,\Omega)} \in \Theta_{(i,j,\Omega)}$. $max\_jump(\ \Psi_{(i,j,n,\Omega)}\ )$ is the maximum jump in path $\Psi_{(i,j,n,\Omega)}$.*

*The Min-Max-Jump distance between a pair of points $i, j$, which belong to $\Omega$, is defined as:*

Table 1: Table of notations

| | |
|---|---|
| $\Omega$ | A set of N points, with each point indexed from 1 to N; |
| $\Omega_{[1,n]}$ | The first $n$ points of $\Omega$, indexed from 1 to n; |
| $\Omega_{n+1}$ | The $(n+1)$th point of $\Omega$; |
| $C_i$ | A cluster of points that is a subset of $\Omega$; |
| $\xi_i$ | One-SCOM of $C_i$; |
| $\Omega + p$ | Set $\Omega$ plus one new point $p$. Since $p \notin \Omega$, if $\Omega$ has N points, this new set now has $N+1$ points; |
| $\Psi_{(i,j,n,\Omega)}$ | $\Psi_{(i,j,n,\Omega)}$ is a sequence from point i to point j, which has length of n points. All the points in the sequence must belong to set $\Omega$. That is to say, it is a path starts from i, and ends with j. For convenience, the path is not allowed to have loops, unless the start and the end is the same point; |
| $d(i,j)$ | $d(i,j)$ is a distance metric between pair of points i and j, such as Euclidean distance; |
| $max\_jump(\Psi_{(i,j,n,\Omega)})$ | $max\_jump(\Psi_{(i,j,n,\Omega)})$ is the maximum jump in path $\Psi_{(i,j,n,\Omega)}$. A jump is the distance from two consecutive points p and q in the path; |
| $\Theta_{(i,j,\Omega)}$ | $\Theta_{(i,j,\Omega)}$ is the set of all paths from point i to point j. A path in $\Theta_{(i,j,\Omega)}$ can have arbitrary number of points (at least two). All the points in a path must belong to set $\Omega$; |
| $MMJ(i,j \mid \Omega)$ | $MMJ(i,j \mid \Omega)$ is the MMJ distance between point i and j, where $\Omega$ is the **Context** of the MMJ distance; |
| $\mathbb{M}_{k,\Omega_{[1,k]}}$ | $\mathbb{M}_{k,\Omega_{[1,k]}}$ is the pairwise MMJ distance matrix of $\Omega_{[1,k]}$, which has shape $k \times k$. The MMJ distances are under the **Context** of $\Omega_{[1,k]}$; |
| $\mathbb{M}_{\Omega}$ | The pairwise MMJ distance matrix of $\Omega$, $\mathbb{M}_{\Omega} = \mathbb{M}_{N,\Omega_{[1,N]}}$; |

$$\Pi = \{max\_jump(\epsilon) \mid \epsilon \in \Theta_{(i,j,\Omega)}\} \tag{1}$$

$$MMJ(i,j \mid \Omega) = min(\Pi) \tag{2}$$

*Where $\epsilon$ is a path from point i to point j, $max\_jump(\epsilon)$ is the maximum jump in path $\epsilon$. $\Pi$ is the set of all maximum jumps. $min(\Pi)$ is the minimum of Set $\Pi$.*

*Set $\Omega$ is called the **Context** of the Min-Max-Jump distance. It is easy to check $MMJ(i,i \mid \Omega) = 0$.*

In summary, Min-Max-Jump distance is the minimum of maximum jumps of all path between a pair of points, under the **Context** of a set of points.

Similar distances have actually been studied in many places in the literature, including the maximum capacity path problem, the widest path problem, the bottleneck edge query problem, the minimax path problem, the bottleneck shortest path problem, and the longest-leg path distance (LLPD) (20; 21; 22; 23).

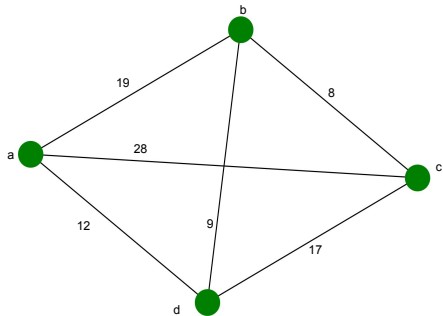

Figure 1: An example

There is a minor difference between Min-Max-Jump distance and other similar distances: Min-Max-Jump distance stresses the *context* of the distance. The *context* is like the condition in conditional probability. The difference becomes non-trivial when we need to calculate the pairwise MMJ distance matrix of a set $S$, under the context of its superset $X$, such as in Section 6.3 of (24). A set $\Omega$ is a superset of another set $B$ if all elements of the set $B$ are elements of the set $\Omega$.

## 3.1  An example

Suppose Set $\Omega$ is composed of the four points in Figure 1. There are five (non-looped) paths from point $a$ to point $c$ in Figure 1:

1. $a \rightarrow c$, the maximum jump is 28;

2. $a \rightarrow b \rightarrow c$, the maximum jump is 19;

3. $a \rightarrow d \rightarrow c$, the maximum jump is 17;

4. $a \rightarrow b \rightarrow d \rightarrow c$, the maximum jump is 19;

5. $a \rightarrow d \rightarrow b \rightarrow c$, the maximum jump is 12.

According to Definition 1, $MMJ(a, c \mid \Omega) = 12$.

To understand Min-Max-Jump distance, imagine someone is traveling by jumping in $\Omega$. Suppose $MMJ(i, j \mid \Omega) = \delta$. If the person wants to reach $j$ from $i$, she must have the ability of jumping at least $\delta$. Otherwise, $j$ is unreachable from $i$ for her. Whether the distance to a point is "far" or "near" is measured by how far (or how high) it requires a person to jump. If the requirement is large, then the point is "far", otherwise, it is "near."

## 3.2  Properties of MMJ distance

**Theorem 1.** *Suppose $i, j, p, q \in \Omega$,*

$$MMJ(i, j \mid \Omega) = \delta \tag{3}$$

$$d(i, p) < \delta \tag{4}$$

$$d(j, q) < \delta \tag{5}$$

*then,*

$$MMJ(p, q \mid \Omega) = \delta \tag{6}$$

where d(x,y) is a distance function (Table 1).

*Proof.* $MMJ(i, j \mid \Omega) = \delta$ is equivalent to $\exists P \in \Theta_{(i,j,\Omega)}$, such that $M(P) = \delta$, and $\forall T \in \Theta_{(i,j,\Omega)}$, $M(T) \geq \delta$, where $\Theta_{(i,j,\Omega)}$ is the set of all paths from point $i$ to point $j$ under context $\Omega$. $M(P)$ is the maximum jump in path $P$. We can assume $MMJ(p, q \mid \Omega) > \delta$ and $MMJ(p, q \mid \Omega) < \delta$, then we will arrive to a contradiction in both cases. □

**Theorem 2.** *Suppose $r \in \{1, 2, \ldots, n\}$,*

$$f(t) = max(d(\Omega_{n+1}, \Omega_t),\ MMJ(\Omega_t, \Omega_r \mid \Omega_{[1,n]})) \tag{7}$$

$$\mathbb{X} = \{f(t) \mid t \in \{1, 2, \ldots, n\}\} \tag{8}$$

*then,*

$$MMJ(\Omega_{n+1}, \Omega_r \mid \Omega_{[1,n+1]}) = min(\mathbb{X}) \tag{9}$$

For the meaning of $\Omega_t, \Omega_r, \Omega_{[1,n]}, and\ \Omega_{[1,n+1]}$, see Table 1.

*Proof.* There are $n$ possibilities of the MMJ path from $\Omega_{n+1}$ to $\Omega_r$, under the context of $\Omega_{[1,n+1]}$, set $\mathbb{X}$ enumerate them all. Each element of $\mathbb{X}$ is the maximum jump of each possibility. Therefore, according to the definition of MMJ distance, $MMJ(\Omega_{n+1}, \Omega_r \mid \Omega_{[1,n+1]}) = min(\mathbb{X})$. □

**Corollary 1.** *Suppose $r \in \{1, 2, \ldots, N\}, p \notin \Omega$,*

$$f(t) = max(d(p, \Omega_t),\ MMJ(\Omega_t, \Omega_r \mid \Omega)) \tag{10}$$

$$\mathbb{X} = \{f(t) \mid t \in \{1, 2, \ldots, N\}\} \tag{11}$$

*then,*

$$MMJ(p, \Omega_r \mid \Omega + p) = min(\mathbb{X}) \tag{12}$$

For the meaning of $\Omega + p$, see Table 1.

*Proof.* The proof follows the conclusion of Theorem 2. □

**Theorem 3.** *Suppose $i, j \in \{1, 2, \ldots, n\}$,*

$$x_1 = MMJ(\Omega_i, \Omega_j \mid \Omega_{[1,n]}) \tag{13}$$

$$t_1 = MMJ(\Omega_{n+1}, \Omega_i \mid \Omega_{[1,n+1]}) \tag{14}$$

$$t_2 = MMJ(\Omega_{n+1}, \Omega_j \mid \Omega_{[1,n+1]}) \tag{15}$$

$$x_2 = max(t_1,\ t_2) \tag{16}$$

*then,*

$$MMJ(\Omega_i, \Omega_j \mid \Omega_{[1,n+1]}) = min(x_1,\ x_2) \tag{17}$$

*Proof.* There are two possibilities of the MMJ path from $\Omega_i$ to $\Omega_j$, under the context of $\Omega_{[1,n+1]}$: $\Omega_{n+1}$ is in the path or it is not in the path. $x_2$ is the min-max jump of the first possibility; $x_1$ is the min-max jump of the second possibility. Therefore, according to the definition of MMJ distance, $MMJ(\Omega_i, \Omega_j \mid \Omega_{[1,n+1]}) = min(x_1,\ x_2)$. □

# 4 Calculation of Min-Max-Jump distance

We propose two methods to calculate the pairwise Min-Max-Jump distance matrix of a dataset. There are other methods for calculating or estimating it, such as a modified SLINK algorithm (25), or with Cartesian trees (26; 27), or from a sequence of nearest neighbor graphs (23), or a modified version of the Floyd–Warshall algorithm.

## 4.1 MMJ distance by recursion

The first method calculates $\mathbb{M}_\Omega$ by recursion. $\mathbb{M}_\Omega$ is the pairwise MMJ distance matrix of $\Omega$ (Table 1). $\mathbb{M}_{k,\Omega_{[1,k]}}$ is the MMJ distance matrix of the first $k$ points of $\Omega$ (Table 1). Note $\mathbb{M}_{2,\Omega_{[1,2]}}$ is simple to calculate. $\mathbb{M}_\Omega = \mathbb{M}_{N,\Omega_{[1,N]}}$. $\mathbb{M}_\Omega$ is a $N \times N$ symmetric matrix. Rows and columns of $\mathbb{M}_\Omega$ are indexed from 1 to N.

Step 7 of Algorithm 1 can be calculated with the conclusion of Theorem 2; Step 12 of Algorithm 1 can be calculated with the conclusion of Theorem 3.

Algorithm 1 has complexity of $\mathcal{O}(n^3)$, where $n$ is the cardinality of Set $\Omega$.

---

**Algorithm 1** MMJ distance by recursion

---

**Input:** $\Omega$
**Output:** $\mathbb{M}_\Omega$

1: **function** MMJ_BY_RECURSION($\Omega$)
2:     $N \leftarrow length(\Omega)$
3:     Initialize $\mathbb{M}_\Omega$ with zeros
4:     Calculate $\mathbb{M}_{2,\Omega_{[1,2]}}$, fill in $\mathbb{M}_\Omega[1,2]$ and $\mathbb{M}_\Omega[2,1]$
5:     **for** $n \leftarrow 3$ to $N$ **do**
6:         **for** $r \leftarrow 1$ to $n-1$ **do**
7:             Calculate $MMJ(\Omega_n, \Omega_r \mid \Omega_{[1,n]})$, fill in $\mathbb{M}_\Omega[n,r]$ and $\mathbb{M}_\Omega[r,n]$
8:         **end for**
9:         **for** $i \leftarrow 1$ to $n-1$ **do**
10:             **for** $j \leftarrow 1$ to $n-1$ **do**
11:                 **if** $i < j$ **then**
12:                     Calculate $MMJ(\Omega_i, \Omega_j \mid \Omega_{[1,n]})$, update $\mathbb{M}_\Omega[i,j]$ and $\mathbb{M}_\Omega[j,i]$
13:                 **end if**
14:             **end for**
15:         **end for**
16:     **end for**
17:     **return** $\mathbb{M}_\Omega$
18: **end function**

---

## 4.2 MMJ distance by calculation and copy

According to the conclusion of Theorem 1, there are many duplicated values in $\mathbb{M}_\Omega$. So in the second method we can calculate the MMJ distance value in one position and copy it to other positions in $\mathbb{M}_\Omega$.

A well-known fact about MMJ distance is: "the path between any two nodes in a minimum spanning tree (MST) is a minimax path." A minimax path in an undirected graph is a path between two vertices $v, w$ that minimizes the maximum weight of the edges on the path. That is to say, it is a MMJ path. By utilizing this fact, we propose Algorithm 2.

---

**Algorithm 2** MMJ distance by Calculation and Copy

---

**Input:** $\Omega$
**Output:** $\mathbb{M}_\Omega$

1: **function** MMJ_CALCULATION_AND_COPY($\Omega$)
2:     Initialize $\mathbb{M}_\Omega$ with zeros
3:     Construct a MST of $\Omega$, noted $T$
4:     Sort edges of $T$ from large to small, generate a list, noted $L$
5:     **for** e in $L$ **do**
6:         Remove $e$ from $T$. It will result in two connected sub-trees, $T_1$ and $T_2$;
7:         Traverse $T_1$ and $T_2$;
8:         For all pair of nodes $(p,q)$, where $p \in T_1$, $q \in T_2$. Fill in $\mathbb{M}_\Omega[p,q]$ and $\mathbb{M}_\Omega[q,p]$ with the weight of $e$.
9:     **end for**
10:     **return** $\mathbb{M}_\Omega$
11: **end function**

---

The complexity of Algorithm 2 is $\mathcal{O}(n^2)$. Because the construction of a MST of a complete graph is $\mathcal{O}(n^2)$. During the "for" part (Step 5 to 9) of the algorithm, it accesses each cell of $\mathbb{M}_\Omega$ only once. Unlike Algorithm 1, which accesses each cell of $\mathbb{M}_\Omega$ for $\mathcal{O}(n)$ times. The merit of the "Calculation and Copy" method is that it is easier to understand than using the Cartesian trees (26; 27).

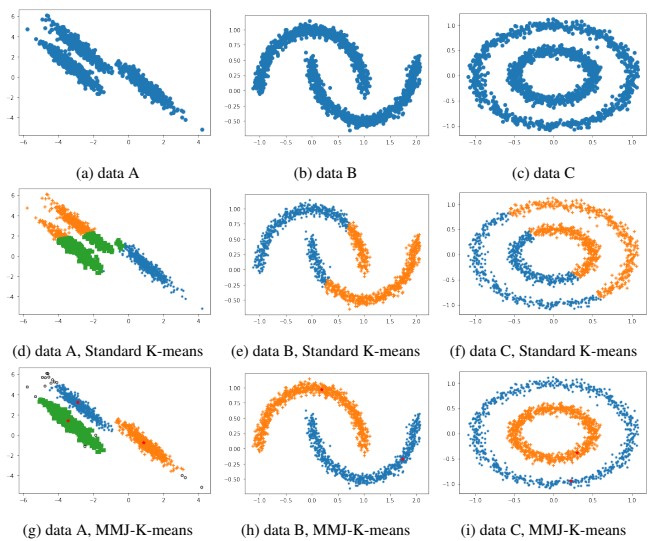

Figure 2: Standard K-means vs. MMJ-K-means

## 5   Applications of Min-Max-Jump distance

We explore two applications of MMJ distance, and test the applications with experiments. All the MMJ distances in the experiments are calculated with Algorithm 1.

### 5.1   MMJ-based K-means

K-means clustering aims to partition $n$ observations into $k$ clusters in which each observation belongs to the cluster with the nearest mean (cluster center or centroid), serving as a prototype of the cluster (28). Standard K-means uses Euclidean distance. We can revise K-means to use Min-Max-Jump distance, with the cluster centroid replaced by the Semantic Center of Mass (SCOM) (particularly, One-SCOM) of each cluster. For the definition of SCOM, see a previous paper (29). One-SCOM is like medoid, but has some difference from medoid. Section 6.3 of (29) compares One-SCOM and medoid. In simple terms, the One-SCOM of a set of points, is the point which has the smallest sum of squared distances to all points in the set.

Standard K-means usually cannot deal with non-spherical shaped data, such as the ones in Figure 2. MMJ-based K-means (MMJ-K-means) can cluster such irregularly shaped data. Figure 2 compares Standard K-means and MMJ-K-means, on clustering three data which come from the scikit-learn project (30). Figure 3 are eight more samples of MMJ-K-means. The data sources corresponding to the data IDs can be found at this URL (temporarily hidden for double blind review).

It can be seen MMJ-K-means can (almost) work properly for clustering the 11 data, which have different kinds of shapes. The black circles are Border points (Definition 2), the red stars are the center (One-SCOM) of each cluster. During training of MMJ-K-means, the Border points are randomly allocated to one of its nearest centers.

**Definition 2.** *Border point*

*A point is defined to be a Border point if its nearest mean (center, centroid, or One-SCOM) is not unique.*

Compared with other clustering models that can handle irregularly shaped data, such as Spectral clustering or the Density-Based Spatial Clustering of Applications with Noise (DBSCAN), the merit of MMJ-K-means is its simplicity; the logic of MMJ-K-means is as simple as K-means. We just replace the Euclidean distance with MMJ distance, and the centroid with the Semantic Center of Mass (SCOM).

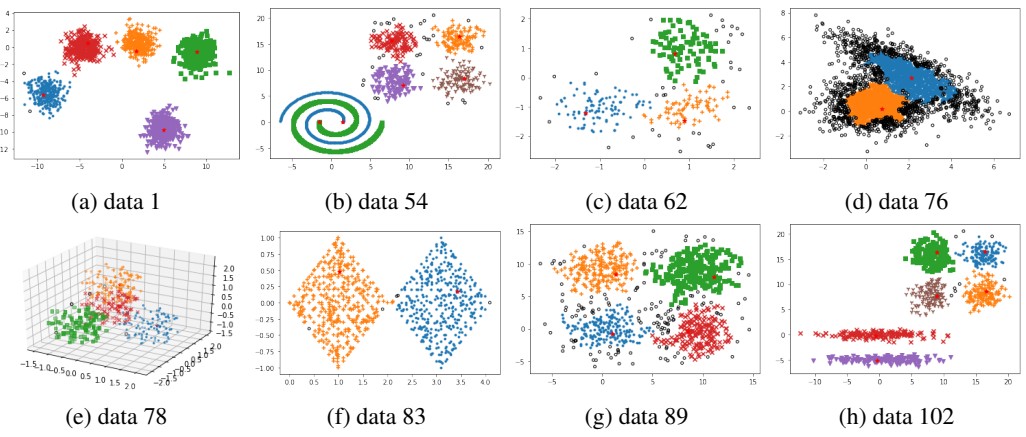

Figure 3: Eight more samples of MMJ-K-means

| | CH | SC | DB | CDbw | DBCV | VIASCKDE | New | MMJ-SC | MMJ-CH | MMJ-DB |
|---|---|---|---|---|---|---|---|---|---|---|
| Accuracy | 27/145 | 38/145 | 42/145 | 8/145 | 56/145 | 11/145 | 74/145 | 83/145 | 90/145 | 69/145 |

Table 2: Accuracy of the ten indices

## 5.2 MMJ-based internal clustering evaluation index

Calinski-Harabasz index, Silhouette coefficient, and Davies-Bouldin index are three of the most popular techniques for internal clustering evaluation. They are used to calculate the goodness of a clustering technique.

The Silhouette coefficient for a single sample is given as:

$$s = \frac{b - a}{max(a, b)}$$

where $a$ is the mean distance between a sample and all other points in the same class. $b$ is the mean distance between a sample and all other points in the next nearest cluster. The Silhouette coefficient for a set of samples is given as the mean of Silhouette coefficient for each sample.

We can also revise Silhouette coefficient to use Min-Max-Jump distance, forming a new internal clustering evaluation index called MMJ-based Silhouette coefficient (MMJ-SC). We tested the performance of MMJ-SC with the 145 datasets mentioned in another paper(31). MMJ-SC obtained a good performance score compared with the other seven internal clustering evaluation indices mentioned in the paper(31). Readers can check Table 2 and compare with Table 5 of Liu's paper(31).

MMJ-based Calinski-Harabasz index (MMJ-CH) and MMJ-based Davies-Bouldin index (MMJ-DB) were also tested. In calculation of these two indices, besides using MMJ distance, the center/centroid of a cluster is replaced by the One-SCOM of the cluster again, as in MMJ-K-means. It can be seen that MMJ distance systematically improves the three internal clustering evaluation indices (Table 2). The best performer is MMJ-CH, which achieves an accuracy of $90/145$. The accuracy of an index is computed by evaluating the index's ability of recognizing the best partition of a dataset from hundreds of candidate partitions(31).

### 5.2.1 Using MMJ-SC in CNNI

The Clustering with Neural Network and Index (CNNI) model uses a Neural Network to cluster data points. Training of the Neural Network mimics supervised learning, with an internal clustering evaluation index acting as the loss function (24). CNNI with standard Silhouette coefficient as the internal clustering evaluation index, cannot deal with non-flat geometry data, such as data B and data C in Figure 2. MMJ-SC gives CNNI model the capability of processing non-flat geometry data. E.g., Figure 4 is the clustering result and decision boundary of data B by CNNI using MMJ-SC. It uses Neural Network C of the CNNI paper (24). CNNI equipped with MMJ-SC, achieves the

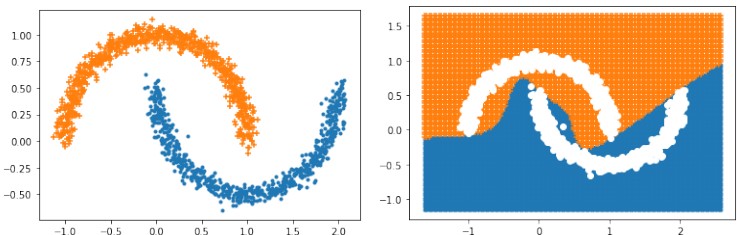

Figure 4: Clustering result and decision boundary of data B by CNNI using MMJ-SC

first inductive clustering model that can deal with non-flat geometry data (24). For the definition of non-flat geometry data, see this[1] Stackexchange question.

# 6 Discussion

## 6.1 Using PAM

Since One-SCOM is like medoid, in MMJ-K-means, we can also use the Partitioning Around Medoids (PAM) algorithm or its variants to find the One-SCOMs (32).

## 6.2 Multiple One-SCOMs in one cluster

There might be multiple One-SCOM points in a cluster, which have the same smallest sum of squared distances to all the points in the cluster. Usually they are not far from each other. We can arbitrarily choose one or keep them all. If we keep them all, then the One-SCOM of a cluster is not a point, but a set of points. If the One-SCOM is a set, when calculating a point's MMJ distance to the One-SCOM of a cluster, we can select the minimum of the point's MMJ distances to all the One-SCOM points.

## 6.3 Differentiating border points

Border points defined in Definition 2 can further be differentiated as weak and strong border points.

**Definition 3.** *Weak Border Point (WBP)*

*A point is defined to be a WBP if its nearest mean (center or One-SCOM) is not unique but less than $K$, where $K$ is the number of clusters.*

**Definition 4.** *Strong Border Point (SBP)*

*A point is defined to be a SBP if its nearest mean (center or One-SCOM) is not unique and equals $K$, where $K$ is the number of clusters.*

Then we can process different kinds of border points with different strategies. E.g., deeming the Strong Border Points as outliers and removing them.

# 7 Conclusion and Future Works

We proposed two algorithms for calculating Min-Max-Jump distance (MMJ distance), and tested two applications of it: MMJ-based K-means and MMJ-based internal clustering evaluation index. MMJ-K-means overcomes a big drawback of K-means, improving its ability of clustering, so that it can handle irregularly shaped clusters. We claim MMJ-CH is the SOTA (state-of-the-art) internal clustering evaluation index, which achieves an accuracy of $90/145$. To thoroughly test the internal clustering evaluation indices, we conducted an experiment on a set of 145 datasets. A normal Machine Learning paper usually uses several or dozens of datasets to test their models or algorithms. In summary, MMJ distance has good capability and potentiality in Machine Learning. Further research may test its applications in other models, such as other clustering evaluation indices.

---

[1] https://datascience.stackexchange.com/questions/52260/terminology-flat-geometry-in-the-context-of-clustering

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
