# OpenReview forum: "Two applications of Min-Max-Jump distance"
_NeurIPS.cc/2024/Conference — Submitted to NeurIPS 2024_

### Official Review · Reviewer_VBcM · 2024-07-09

**Soundness:** 2
**Presentation:** 1
**Contribution:** 1
**Rating:** 3
**Confidence:** 4

**Summary:**

The paper proposes to use a new distance, Min-Max-Jump, which is the minimum largest distance on any path between two points, to be used in k-means clustering to learn clusters.

**Strengths:**

The distance can overcomes some demerits of the convex ("spherical" in the paper) clusters.

**Weaknesses:**

The distance in the paper is, in fact, related to single linkage clustering that assign give a pair of points a distance at which the pair is joint to one cluster. This need to be analyzed to relate to previous work as well as to compute pairwise distances efficiently.

Theoretical property of the distance is poor. The paper should review many other density-based distance functions to put this work into the correct context.

There would be a lot of problems using this distance as many of pairs of nodes would share the same distance. There is no analysis on the  metric property of this distance.

On evaluation, the methods need to compare to single-linkage clustering (SSL) at the very least as all the advantages of using this distance with k-means are available in SLL in its simplest form.

Presentation-wise, it is hardly up the standard. There are methods/algorithms/concepts that are mentioned as "a is like b with a difference" without a formal definition. This mixes up definitions and properties.

**Questions:**

N/A

**Limitations:**

The paper uses a new distance without theoretical justifications. It learns nonconvex clusters by using a nonconvex clustering-based distance (without explicitly mentioning it), which is hardly a novelty.

---

> ### Author Rebuttal · Authors · 2024-08-06
>
> Question:
>
> "The distance in the paper is, in fact, related to single linkage clustering that assign give a pair of points a distance at which the pair is joint to one cluster. This need to be analyzed to relate to previous work as well as to compute pairwise distances efficiently."
> "On evaluation, the methods need to compare to single-linkage clustering (SSL) at the very least as all the advantages of using this distance with k-means are available in SLL in its simplest form."
>
> Response:
>
> Thanks for the constructive suggestions. We will revise the paper according to Reviewer VBcM's suggestions.
>
> MMJ-K-means typically outperforms other algorithms in handling irregular clusters. For example, single linkage clustering is very sensitive to outliers, whereas MMJ-K-means is not, as demonstrated in Figure 3, where all datasets contain some outliers.
>
> Density-based clustering algorithms like DBSCAN are highly sensitive to their hyper-parameters. DBSCAN's hyper-parameter "eps,"  defined as "the maximum distance between two samples for one to be considered as in the neighborhood of the other," is not intuitive and difficult to optimize. When parameter "eps" is fixed, adjusting the coordinates of data points by scaling them can lead to different clustering results. That means if we use a different measurement units for measuring distance (e.g., using kilometers instead of meters), we will get a different clustering result, which is undesirable.
>
> Additionally, DBSCAN has an extra hyper-parameter, "$ min\\_samples $," which is "the number of samples (or total weight) in a neighborhood for a point to be considered as a core point." Having more hyper-parameters requires more effort to tune. In contrast, MMJ-K-means has only one hyper-parameter, "K," which represents the number of clusters. The hyper-parameter K is very intuitive.
>
> Comparison between MMJ-K-means and HDBSCAN is shown in the auxiliary PDF file.
>
> Question:
>
> "Theoretical property of the distance is poor. The paper should review many other density-based distance functions to put this work into the correct context."
> "There would be a lot of problems using this distance as many of pairs of nodes would share the same distance. There is no analysis on the metric property of this distance."
>
> Response:
>
> It is true that many of pairs of nodes share the same distance. However, whether this property of the distance is a merit or demerit,  is not confirmed. E.g., the distance works good in MMJ-K-means, MMJ-based internal clustering evaluation indices, and the Clustering with Neural Network and Index (CNNI) model. There is no sign that having many duplicated distance values is a demerit. It is possible that having many duplicated distance values is actually a merit, because the problem becomes simpler.
>
> MMJ distance fulfills all the requirements as a metric space, e.g., the triangle inequality holds for MMJ distance. Due to the page limit, we cannot discuss more about this in the paper.
>
> We will review more density-based distance functions literature according to Reviewer VBcM's suggestion.
>
> Question:
>
> "Presentation-wise, it is hardly up the standard. There are methods/algorithms/concepts that are mentioned as "a is like b with a difference" without a formal definition. This mixes up definitions and properties."
>
> Response:
>
> We will polish the writing according to all the reviewers'  constructive suggestions.  We will employ a professional  Academic Editing Service if the paper is accepted.
>
> The Semantic Center of Mass of Length One (One-SCOM) has been defined in a previous paper, there is no need to re-define it in this paper.
>
>
> Question:
>
> "The paper uses a new distance without theoretical justifications. It learns nonconvex clusters ..., which is hardly a novelty."
>
> Response:
>
> We have provided some theoretical justifications of the properties of the distance. The distance shares some common properties with other similar distances, such as the longest-leg path distance (LLPD), the minimax path problem, and the widest path problem etc. We have cited other similar distances so that readers can have a grasp of other properties of the distance from the literature.
>
> The fundamental contributions of this work include:
>
> 1. Implementing the MMJ-based internal clustering evaluation index within the Clustering with Neural Network and Index (CNNI) model, achieving the first inductive clustering model capable of handling non-flat geometry data. An inductive clustering model not only can calculate the label of a point in the dataset, but also it can calculate the label of a new point that is not in the dataset. Its supremacy is obvious. If you know another clustering model with this capability, please inform us.
>
> 2. Algorithm 2 (MMJ distance by Calculation and Copy) is currently the fastest algorithm for computing the all points path distance (APPD) matrix. It can address the all-pairs minimax path problem or widest path problem in undirected dense graphs. In a recent study, we implemented and tested the algorithm, and experimental results confirm it as the fastest for solving the APPD matrix. The source code is publicly available, but we cannot provide an URL due to the double-blind review process.
>
> 3. As noted by Reviewer vcc5, Algorithm 1 (MMJ distance by recursion) supports online machine learning, where data is sequentially available. The algorithm allows for a warm start, unlike other algorithms for calculating the MMJ distance matrix, which can only be cold-started.
>
> 4. MMJ-CH is the state-of-the-art internal clustering evaluation index, achieving an accuracy of 90/145. We have provided an API for readers to test their own internal clustering evaluation indices. If you find an index that outperforms MMJ-CH, please let us know.
>
> 5. MMJ-K-means has some very good and unique properties. We have compared MMJ-K-means with other popular clustering models like single linkage clustering, DBSCAN, and HDBSCAN in previous response.

---

> > ### Comment · Reviewer_VBcM · 2024-08-13
> >
> > I thank the authors for the rebuttal. As also in the rebuttal, the paper lacks much justifications to be accepted in this conference. I'd like to keep my score.

---

### Official Review · Reviewer_vcc5 · 2024-07-09

**Soundness:** 4
**Presentation:** 1
**Contribution:** 3
**Rating:** 4
**Confidence:** 3

**Summary:**

This paper proposes a new metric, min-max jump distance. Effectively, say we are given a complete graph with vertex set $\Omega$ and edge weights $d(x,y)$ denoting the distance between $x$ and $y$, where $d$ is a metric. Then $MMJ(x,y|\Omega')$ is the minimum, over all paths between $x$ and $y$, of the maximum weight edge between $x$ and $y$ on the subgraph induced on the vertices $\Omega' \subseteq \Omega$. Explained nicely in the paper, if you started at vertex $x$ and wanted to get to $y$ and $d(\cdot,\cdot)$ denoted the distance required to ``jump'' from one vertex to another, what is the minimum distance you need to be able to jump to somehow traverse from $x$ to $y$?

This is a nice intuitive metric, and has strong connections to the minimum spanning tree. In fact, I suspect there is more literature to draw from minimum spanning tree research that could yield conclusions about MMJ. The MST is also nice in clustering since oddly shaped clusters (non-convex, for instance) can have small MSTs. This is the idea of MMJ: use it as a metric for K-means so that it can identify non-spherical clusters.

The paper proves some notable theory about the properties of MMJ. Mostly, they show how: 1) When adding a new vertex $p$ to a set $\Omega$, $p$'s MMJ within the context of $\Omega+p$ can be computed knowing all pairwise MMJ's within $\Omega$ within the context of $\Omega$. This effectively adds a new point and evaluates it within the complete, updated context. 2) Given the MMJs for this additional point $p$ within the context of $\Omega +p$, expand the MMJ context of all other pairs in $\Omega$ to the context of $\Omega+p$.

This can then be used in a very dynamic programming-like manner to start with just two points, add new points $p$ and find the context of $p$ and all other points, and then update all known existing MMJs to the new context. This is their algorithm 1, requiring $O(n^3)$ time. They also use properties of the MST to bypass unnecessary calculation to yield algorithm 2, which takes only $O(n^2)$ time (to find the MST).

They then evaluate the performance of algorithms using the MMJ measure on irregular shaped clusters to verify that MMJ helps identify these. This makes sense, since they likely have small MSTs, but not small average/sum/max/etc distances within clusters. Notably, they show how MMJ improves K-means.

**Strengths:**

I think MMJ is a very cool metric with nice properties and intuition, particularly that related to the MST (I wish the authors had spent more time discussing this!). Their findings are nice and relatively simple to understand (in spite of the presentation). They show that it helps K-means expand to more complicated cluster shapes, and overall it is a very nice, NeurIPS-worthy result. However, as I will explain in the weaknesses section, I do not think this paper is in an acceptable state for NeurIPS.

**Weaknesses:**

There are a few notable downsides. In terms of the result, I'm not entirely convinced of its novelty. How much of this is actually a re-iteration of MST-based methods already understood? Is this really better than other MST-based algorithms on irregular clusters (think single linkage)? I know that there is a lot of literature that explores irregular shaped clusters, but I am not an expert in this area and so I cannot place this work in the context of existing results. I wish the authors would explain that. Though even if these algorithms aren't entirely better than state of the art, the novelty of the nice formulation of MMJ is certainly appreciated.

However, the biggest flaw in the paper is the writing quality. There are places where the paper is nice and concise, but most of the time it just lacks exposition to understand the higher level of things or adequate details to fully understand what is happening. Formal proofs are contained in the paper, but the jumps in some of the proofs are too large. Theorems and proofs are placed back to back with no high level explanation. Algorithms are written and pseudocode with only the briefest justifications, and no thorough explanations. This is not an acceptable paper for NeurIPS, and I think these issues are too extensive to simply ask the authors to revise. Though if other commenters disagree, I am amenable to changing my opinion.

And one of the disappointing things about this is how natural this work is and how much it lends itself to nice intuitive explanations and visual depictions! For instance, you could do some very nice visualizations of Algorithm 1, where you depict a matrix and show which indices have been calculated to what context $\Omega_n$ at each time point. This clarifies the different purpose of the two loops.

I hope to see this paper submitted again later in a more cleaned up state!

**Questions:**

1. What exactly is a path-based distance?

2. When you define MMJ, you do not specify exactly what a path is. All you introduce is a point set and a distance function. Are we to assume that a "path" is done on the complete graph with edge weights equal to the distances between points? Please clarify in the paper. I read the paper with this assumption.

3. How would MMJ compare to the similarly defined max-min jump measure? I would think it would be more akin to a similarity measure than a distance.

4. Line 114: what is M(P)? Also, how do you arrive at the contradiction in this proof? This needs more explanation!

5. Line 122: What do you mean there are n possibilities? Aren't there something like n! candidate paths, since you check all paths of all lengths with a fixed set of endpoints?

6. Thm 2 proof: You say X enumerates them all - what about the direct path that just traverses the single edge between the points?

7. Is Corollary 1 actually any different from Thm 2? It seems like in both you just have a set $\Omega_{[1,n]}$ and then you have a point outside of the set that you want to get the MMJ to (with respect to some reference point). I think Corollary 1 is just a cleaner way to state Theorem 2.

8. In Theorem 3, is it actually true that $x_2$ is the MMJ if $\Omega_{n+1}$ is in the path? It doesn't break your argument, but if the solution path doesn't go through $\Omega_{n+1}$, then it's possible that the MMJ path from i to n+1 and from j to n+1 share an edge. So the MMJ path from i to j that goes through n+1 could actually be worse since we aren't allowed to traverse that edge twice. Again, the proof still works, but I think that claim only holds if the MMJ path from i to j does, in fact, go through n+1. Otherwise, we might have something larger, which is ignored by taking the minimum.

9. Section 4.1: Why does k appear twice in the matrix subscripts? Couldn't it just be something like $M_k$?

10. How does MMJ-K-means perform on spherical data compared to K-means? Presumably, worse. It would be nice to quantify how much worse.

**Limitations:**

None notable

---

> ### Author Rebuttal · Authors · 2024-08-06
>
> Thanks for the detailed and insightful review.
>
> Reviewer vcc5's  main concern is the writing quality. We will try our best to improve the writing quality according to all the reviewers' suggestions. We will employ a professional  Academic Editing Service if the paper is accepted.
>
> We have open-sourced the algorithms and models in this paper. Be debugging step-by-step with a simple example, it is not very difficult for readers to grasp all the details of the algorithms and models discussed in the paper.
>
> Reviewer vcc5 indicated a key merit of Algorithm 1 (MMJ distance by recursion): the algorithm can be warm-started. In other words, the algorithm supports online machine learning. While other algorithms (e.g., a modified version of the Floyd-Warshall algorithm) for solving MMJ distance matrix need to be cold-started. A warm-start can be much faster than a cold-start.
>
> Question:
>
> "How much of this is actually a re-iteration of MST-based methods already understood? Is this really better than other MST-based algorithms on irregular clusters (think single linkage)?"
>
> Response:
>
> MMJ-K-means generally performs better than other algorithms which can deal with irregular clusters. E.g., single linkage clustering is very sensitive to outliers. MMJ-K-means is not sensitive to outliers, this merit is confirmed by Figure 3. In Figure 3, all the datasets have some outliers. Density-based clustering algorithms like DBSCAN are very sensitive to their hyper-parameters. DBSCAN has a hyper-parameter "eps", which is "the maximum distance between two samples for one to be considered as in the neighborhood of the other." The "eps" hyper-parameter is not intuitive and difficult to optimize. If we fix the "eps" hyper-parameter and just magnify or shrink each data point's coordinates, we may get a different clustering result. This is not good.
>
> DBSCAN has an extra hyper-parameter "$ min\\_samples $", which is "the number of samples (or total weight) in a neighborhood for a point to be considered as a core point." More hyper-parameters means we need more effort to adjust it. MMJ-K-means  has only one hyper-parameter, the "K," which is the number of clusters.
>
> Comparison between MMJ-K-means and HDBSCAN is shown in the auxiliary PDF file.
>
>
> Question 1:
>
> "What exactly is a path-based distance?"
>
> Response:
>
> Path-based distance is different from coordinate-based distance. In a coordinate-based distance, such as Euclidean or Manhattan distance, the distance between two points is totally determined by their coordinates. In a path-based distance like MMJ distance, we need to check all the paths between two points in a graph, to calculate their distance.
>
> Question 2:
>
> "When you define MMJ, you do not specify exactly what a path is ..."
>
> Response:
>
> Yes. You are right. A dataset can be straightforwardly converted to a complete graph.
>
> Question 3:
>
> "How would MMJ compare to the similarly defined  ..."
>
> Response:
>
> MMJ distance is a distance, not a similarity measure. It fulfills all the requirements as a metric space, e.g., the triangle inequality holds for MMJ distance. A similarity measure does not need to fulfill the requirements of a metric space.
>
> Question 4:
>
> "Line 114: what is  ..."
>
> Response:
>
> The meaning of M(P) is explained in Line 115 and 116. M(P) is the maximum jump in path P.
> The contradiction can be arrived intuitively with the definition of MMJ distance.  If we assume $ MMJ(p,q ~| ~\Omega) > \delta $, that means if a person what to reach q from p, she must have the ability of jumping farther than $ \delta $. If the person only has a jumping ability of $ \delta $, then q is unreachable from p for her. However, the person is a really smart girl, she choose to jump from p to i, then i to j along an MMJ path, then j to q. Amazingly, she just reached q from p with a jumping ability of $ \delta $. This is the contradiction. Another contradiction can be arrived similarly. The explanation is wordy, however, the logic is simple.
>
> Question 5:
>
> "Line 122: What do you mean there are  ..."
>
> Response:
>
> The $ n! $ candidate paths can be classified into n possibilities. If a person starts jumping from $ \Omega_{n+1} $, how many choices are there for the person to choose as the second point? There are  only n choices, $ \Omega_1 $, $ \Omega_2 $, ..., $ \Omega_n $. After the second point in the path is determined, then the person can jump arbitrarily in  context $ \Omega_{[1,n]}$. We already know all the MMJ distances under context $ \Omega_{[1,n]}$. Note the paths are loop-less, $ \Omega_{n+1} $ appears as the first point in the path, it will not appear in the remaining points of the path.
>
> Question 6:
>
> "Thm 2 proof: You say X enumerates  ..."
>
> Response:
>
> Good question! The direct path that just traverses the single edge between the points is included in the n possibilities. In this case, the second item in Equation 7 is just zero. Note a point's MMJ distance to itself is always zero, no matter under what context.
>
> Question 7:
>
> "Is Corollary 1 actually any different from  ..."
>
> Response:
>
> Good  point. Corollary 1 only has some notation difference from Theorem 2. It is indeed easier to understand than Theorem 2. However, without Theorem 2, it is harder to explain Step 7 of Algorithm 1 to readers.
>
> Question 8:
>
> "In Theorem 3, is it actually  ..."
>
> Response:
>
> Yes. If $ \Omega_{n+1} $ is confirmed in the path. Then $ x_2 $ is the MMJ distance.
>
> Question 9:
>
> "Section 4.1: Why does k appear twice  ..."
>
> Response:
>
> The second k indicates the context of the MMJ distance matrix. There could be multiple choices for the second k, even when the first k is fixed. E.g., the second k can be set to k, k+1, k+2, ..., N.
>
> Question 10:
>
> "How does MMJ-K-means perform  ..."
>
> Response:
>
> MMJ-K-means performs as good as K-means on spherical data. See data 1, data 62, data 78, and data 89 in Figure 3, they are spherical data.

---

> > ### Comment · Reviewer_vcc5 · 2024-08-12
> >
> > I appreciate the authors' responses, however the writing quality issue still stands and I believe the changes required are too great to allow acceptance.

---

### Official Review · Reviewer_BgWh · 2024-07-12

**Soundness:** 3
**Presentation:** 2
**Contribution:** 2
**Rating:** 4
**Confidence:** 4

**Summary:**

This paper presents the Min-Max-Jump (MMJ) distance concept and two calculation methods, focusing on path optimization in data analysis and clustering. The contributions include introducing MMJ distance, proposing efficient calculation methods, discussing its properties and applications, and offering a user-friendly approach for practical implementation. Overall, the paper introduces a new distance metric for path optimization and data analysis, providing useful tools and insights for various applications in the field.

**Strengths:**

S1: The paper demonstrates strength through its meticulous use of theorems and proofs, enhancing the credibility and robustness of the research findings.

S2: Clear visualization of results in the paper aids in effectively conveying complex information to the readers, improving understanding and interpretation.

S3: Extensive literature citations throughout the paper showcase a strong foundation of existing knowledge and research, adding depth and scholarly rigor to the study's presentation.

**Weaknesses:**

W1: The paper's writing style deviates from academic norms, indicating a need for improvement in writing proficiency.

W2: The extremely brief Introduction lacks a detailed definition of the problem, its significance, and challenges. Moreover, it lacks citation support for the points presented. While Section 2.1 mentions methods like k-NN, UMAP, HDBSCAN, it fails to provide corresponding references, lacking essential academic backing.

W3: The overall structure of the paper lacks clarity, as it introduces different distance metrics in Section 2.1 but introduces a new distance measurement approach in Section 2.4, leading to disjointed logic.

W4: The presentation of various distance metrics in Section 2.1 appears disorganized and lacks coherence.

W5: The extensive definition provided towards the end of Section 6.3 disrupts the logical flow of the paper, suggesting a need to adjust the paper's structural coherence.

**Questions:**

Q1:Can the authors provide more insights into the practical implications of their research findings and how they can be applied in real-world scenarios?

**Limitations:**

The paper does not discuss limitations. The authors seem to perceive their work as solely testing models with datasets without considering the shortcomings of the algorithms themselves.

---

> ### Author Rebuttal · Authors · 2024-08-06
>
> Reviewer BgWh's main concern is the writing style of the paper. We will polish the writing according to Reviewer BgWh's  constructive suggestions. We will employ a professional  Academic Editing Service if the paper is accepted.
> 1. We will revise the introduction section to present a detailed definition of the problem, its significance, and challenges.
> 2. We will revise Section 2.1 to incorporate corresponding references to k-NN, UMAP, HDBSCAN.
> 3. We will refine Section 2.1, and merge Section 2.1 with Section 2.4, to improve the structural coherence.
> 4. We will move the definitions in Section 6.3 to previous sections.
>
> Question:
>
> "Q1:Can the authors provide more insights into the practical implications of their research findings and how they can be applied in real-world scenarios?"
>
> Response:
>
> 1. Both Algorithm 1 and Algorithm 2  can be used to solve the all-pairs minimax path problem or widest path problem. Algorithm 1 (MMJ distance by recursion) supports online machine learning, in which data becomes available in a sequential order. While traditional algorithm like a modified Floyd-Warshall algorithm does not have this merit. A new study shows Algorithm 2 (MMJ distance by Calculation and Copy) is the fastest algorithm for solving the all points path distance (APPD) matrix by far (see our response to Reviewer p2N6). Solving the minimax path problem or widest path problem has extensive practical applications in fields like Network Routing, Transportation, Supply Chain Management, and Telecommunication Networks etc.
>
> 2. MMJ-K-means model overcomes a  key drawback of  standard K-means model. It can be used as a replacement of K-means model on clustering data, especially when the datasets that are not the union of well-separated, spherical clusters, where standard K-means is unusable in these settings. For real-world large high dimensional data like ImageNet, we can pre-process the data with dimension reduction or representation learning techniques, then use MMJ-K-means and MMJ-based internal clustering evaluation index to analyze the processed data.
>
> Question:
>
> "The paper does not discuss limitations. The authors seem to perceive their work as solely testing models with datasets without considering the shortcomings of the algorithms themselves."
>
> Response:
>
> A possible limitation of the models is that the models may not work in high dimensional. However, due to "curse of dimensionality," a low-dimensional model does not necessarily need to work in high dimensional, it is acceptable that it only works in  low-dimensional settings. A low-dimensional model can still be used to analyze high dimensional data. We just need to pre-process the data with dimension reduction or representation learning techniques, then use the low-dimensional model to analyze the processed data.

---

### Official Review · Reviewer_p2N6 · 2024-07-12

**Soundness:** 3
**Presentation:** 2
**Contribution:** 2
**Rating:** 4
**Confidence:** 5

**Summary:**

Different distance metrics have been introduced in the literature for data analysis. In this paper the authors consider the min-max-jump distance and apply it in the context two applications, namely, k-means clustering and as an internal clustering evaluation index. They also present two algorithms for computing the min-max-jump distance.

Experimental comparisons reveal that min-max-jump based k-means clustering is better than standard k-means clustering. Also, the min-max-jump distance is shown to be a better internal clustering evaluation index.

This referee feels that this work is rather incremental. Also, experiments have been conducted only on very small datasets.

**Strengths:**

The authors demonstrate the efficacy of the min-max-jump distance on two different applications.

Experimental results have also been supplied to support their assertions.

**Weaknesses:**

The work done is incremental with very limited novelty.
Extensive experiments are called for.

**Questions:**

None

---

> ### Author Rebuttal · Authors · 2024-08-06
>
> Question:
>
> "This referee feels that this work is rather incremental." "The work done is incremental with very limited novelty."
>
> Response:
>
> The research is not incremental, but fundamental. The fundamental contributions of the work include:
>
> 1. Applying MMJ-based internal clustering evaluation index to the Clustering with Neural Network and Index (CNNI) model, achieves the first inductive clustering model that can deal with non-flat geometry data. If you know another clustering model has this property, please let us know.
>
> 2. Algorithm 2 (MMJ distance by Calculation and Copy) is the fastest algorithm for solving the all points path distance (APPD) matrix by far. It can be used to solve the all-pairs minimax path problem or widest path problem, in undirected dense graphs. In a new study, we have implemented and tested the algorithm. Experiment result shows it is the fastest algorithm for solving the APPD matrix by far. The source code of the algorithm is now publicly accessible. However, we cannot provide an URL of it, because of double-blind review.
>
> 3. As indicated by  Reviewer vcc5, Algorithm 1 (MMJ distance by recursion) supports online machine learning, in which data becomes available in a sequential order. The algorithm has a merit of warm-start. While other algorithms for solving the APPD matrix can only be cold-started.
>
> 4. MMJ-CH is the SOTA (state-of-the-art) internal clustering evaluation index, which achieves an accuracy of 90/145. We have left an API for readers to test their own internal clustering evaluation index. If you have found an index that performs better than MMJ-CH, please let us know.
>
> 5. A key drawback of K-means is that it cannot deal with datasets that are not the union of well-separated, spherical clusters. MMJ-K-means has overcome this demerit of K-means. MMJ-K-means generally outperforms other algorithms which can deal with irregular clusters.
>
>
> Question:
>
> "Also, experiments have been conducted only on very small datasets."
>
> Response:
>
> We have not tested the models on large high dimensional data like ImageNet, because the models introduced in the paper are supposed to be low-dimensional models. Due to "curse of dimensionality," a low-dimensional model does not necessarily need to work in high dimensional, it is acceptable that it only works in  low-dimensional settings. A low-dimensional model can still be used to analyze high dimensional data, by pre-processing the data with dimension reduction or representation learning techniques, then use the low-dimensional model to analyze the processed data.
>
> Another reason for not testing the models on large high dimensional dataset is that there is no trustworthy ground-truth labeling  in high dimensional data. Almost all of the high dimensional datasets are labeled by some humans, there inevitably
> exists subjectivity in labeling high dimensional data points. E.g., if we invite a different group of humans to re-label the MNIST or ImageNet dataset, it is highly possible that we will obtain a different ground-truth labeling of the datasets. Therefore, the ground-truth of high dimensional datasets are actually not real ground-truth. In low dimensional like 2 or 3 dimension, we can understand the dataset by directly observing the layout of the dataset, there is no subjectivity involved when the clusters are well-separated, so the ground-truth is real ground-truth in low dimensions. Even when the clusters are overlapping with each other, we can directly observe how the clusters are overlapped. In high dimensional, we cannot do this.
>
> To test the MMJ-based models with large high dimensional data, we need a  trustworthy ground-truth labeling to be compared with. To be a  trustworthy ground-truth, the labeling should be objective, it should not be decided by some humans. Unfortunately, there is no such trustworthy ground-truth labeling  for high dimensional data.
>
>
> Question:
>
> "Extensive experiments are called for."
>
> Response:
>
> We have tested the models and algorithms using 145 clustering benchmark datasets. Typically, a standard machine learning paper tests its models or algorithms on several or at most dozens of datasets. In contrast, our study uses hundreds of datasets, making our experiments exceptionally extensive.

---

> > ### Comment · Reviewer_p2N6 · 2024-08-12
> >
> > Rebuttal has been read.

---

### Author Rebuttal · Authors · 2024-08-06

Thanks for all the reviewers' insightful review and constructive suggestions.

 The fundamental contributions of the work can be summarized as following:

1. Applying MMJ-based internal clustering evaluation index to the Clustering with Neural Network and Index (CNNI) model, achieves the first inductive clustering model that can deal with non-flat geometry data. An inductive clustering model not only can calculate the label of a point in the dataset, but also it can calculate the label of a new point that has not seen. Its supremacy is obvious. There is no other clustering model has this merit.

2. Algorithm 2 (MMJ distance by Calculation and Copy) is the fastest algorithm for solving the all points path distance (APPD) matrix by far. It can be used to solve the all-pairs minimax path problem or widest path problem, in undirected dense graphs. In a new study, we have implemented and tested the algorithm. The experiment is conducted on an ordinary desktop computer with "3.3 GHz Quad-Core Intel Core i5" CPU and 16 GB RAM. The experiment result shows it is the fastest algorithm for solving the APPD matrix by far. It can calculate the APPD matrix of 10,000 points in about 67 seconds, while other algorithms cannot finish the calculation in two hours. The algorithm is now implemented with Python. If it is implemented with C/C++, it can be even faster. The source code of the algorithm is now publicly accessible. However, we cannot provide an URL of it, because of double-blind review.

3. As indicated by  Reviewer vcc5, Algorithm 1 (MMJ distance by recursion) supports online machine learning, in which data becomes available in a sequential order. The algorithm has a merit of warm-start. While other algorithms for solving the all points path distance (APPD) matrix can only be cold-started. As we all known, a warm-start can be much faster than a cold-start. E.g., suppose we have calculated the APPD matrix $ M_G $ of a large graph $ G $, then we got a new point (or node) $ p $,  where $ p \notin G$. The new graph is noted $G + p$. To calculate the APPD matrix of graph $G + p$, if we use other algorithms, we need to start from zero. Algorithm 1 has the merit of utilizing the calculated $ M_G $ for computing the new APPD matrix, with the conclusions of the theorems and corollary introduced in this paper. This is especially useful when the graph is a directed dense graph, where starting from zero needs $ O(n^3) $ complexity, but a warm-start of Algorithm 1 (MMJ distance by recursion) only needs $ O(n^2) $ complexity.

4. MMJ-CH is the SOTA (state-of-the-art) internal clustering evaluation index, which achieves an accuracy of 90/145. We have left an API for readers to test their own internal clustering evaluation index. If you have found an index that outperforms MMJ-CH, please let us know.

5. A key drawback of K-means is that it cannot deal with datasets that are not the union of well-separated, spherical clusters. MMJ-K-means has overcome this demerit of K-means.

MMJ-K-means generally performs better than other algorithms which can deal with irregular clusters. E.g., single linkage clustering is very sensitive to outliers. MMJ-K-means is not sensitive to outliers, this merit is confirmed by Figure 3. In Figure 3, all the datasets have some outliers.

Density-based clustering algorithms like DBSCAN are very sensitive to their hyper-parameters. DBSCAN has a hyper-parameter "eps," which is "the maximum distance between two samples for one to be considered as in the neighborhood of the other." The "eps" hyper-parameter is not intuitive and difficult to optimize. If we fix the "eps" hyper-parameter and just magnify or shrink each data point's coordinates, we may get a different clustering result. That means if we use a different measurement units for measuring distance (e.g., using kilometers instead of meters), we will get a different clustering result, which is undesirable.

DBSCAN has an extra hyper-parameter "$ min\\_samples $," which is "the number of samples (or total weight) in a neighborhood for a point to be considered as a core point." More hyper-parameters means we need more effort to adjust it. MMJ-K-means  has only one hyper-parameter, the "K," which is the number of clusters. The hyper-parameter is very intuitive. If we fix hyper-parameter K in MMJ-K-means, then adjust the coordinates of data points by scaling them,  the clustering result keeps invariant.

MMJ-K-means also performs better than HDBSCAN. Comparison between MMJ-K-means and HDBSCAN is shown in the auxiliary PDF file.

The models introduced in the paper are supposed to be low-dimensional models. A possible limitation of the paper is that the introduced models may not work in high dimensional. However, due to "curse of dimensionality," a low-dimensional model does not necessarily need to work in high dimensional, it is acceptable that it only works in  low-dimensional settings. A low-dimensional model can still be used to analyze high dimensional data. We just need to pre-process the data with dimension reduction or representation learning techniques, then use the low-dimensional model to analyze the processed data.

The reviewers have concern about the writing of the paper. We will try our best to improve the writing quality according to all the reviewers' suggestions. We will employ a professional  Academic Editing Service if the paper is accepted.

---

### Decision · Program_Chairs · 2024-09-25

**Decision:**

Reject

**Comment:**

The four reviewers are well-qualified and diverse, so the fact that none recommend acceptance is dispositive. The author has responded in detail, but has not changed the mind of any reviewer. Reviewer vcc5 is a relevant expert and is sympathetic. However, s/he still recommends rejection, like the other reviewers.

The author must understand that the writing changes needed are major. Employing an editor to improve the English in a surface way is likely not enough. Instead, the paper needs to follow the expository conventions of modern ML research, with more comprehensive, more precise, and more thoughtful qualitative discussion. For example, it is not sufficient simply to cite a web page for the meaning of the term "non-flat." And many previous papers have provided algorithms for clustering data such as that in Figures 2, 3, and 4, for example Manifold Clustering by Souvenir and Pless in Proceedings of the 10th International Conference on Computer Vision (ICCV 2005), pp. 648 – 653.

It is possible that this paper should be two separate submissions, one that is purely algorithmic for supposedly the fastest algorithm for computing all points path distances, and one for the application of this algorithm to clustering.